# Unprecedentedly high activity and selectivity for hydrogenation of nitroarenes with single atomic Co$_1$-N$_3$P$_1$ sites

Hongqiang Jin[1,2], Peipei Li[1,2], Peixin Cui [3], Jinan Shi[4,5], Wu Zhou [4,5], Xiaohu Yu [6✉], Weiguo Song[1,2] & Changyan Cao [1,2✉]

Transition metal single atom catalysts (SACs) with M$_1$-N$_x$ coordination configuration have shown outstanding activity and selectivity for hydrogenation of nitroarenes. Modulating the atomic coordination structure has emerged as a promising strategy to further improve the catalytic performance. Herein, we report an atomic Co$_1$/NPC catalyst with unsymmetrical single Co$_1$-N$_3$P$_1$ sites that displays unprecedentedly high activity and chemoselectivity for hydrogenation of functionalized nitroarenes. Compared to the most popular Co$_1$-N$_4$ coordination, the electron density of Co atom in Co$_1$-N$_3$P$_1$ is increased, which is more favorable for H$_2$ dissociation as verified by kinetic isotope effect and density functional theory calculation results. In nitrobenzene hydrogenation reaction, the as-synthesized Co$_1$-N$_3$P$_1$ SAC exhibits a turnover frequency of 6560 h$^{-1}$, which is 60-fold higher than that of Co$_1$-N$_4$ SAC and one order of magnitude higher than the state-of-the-art M$_1$-N$_x$-C SACs in literatures. Furthermore, Co$_1$-N$_3$P$_1$ SAC shows superior selectivity (>99%) toward many substituted nitroarenes with co-existence of other sensitive reducible groups. This work is an excellent example of relationship between catalytic performance and the coordination environment of SACs, and offers a potential practical catalyst for aromatic amine synthesis by hydrogenation of nitroarenes.

[1] Beijing National Laboratory for Molecular Sciences, CAS Research/Education Center for Excellence in Molecular Sciences, Laboratory of Molecular Nanostructures and Nanotechnology, Institute of Chemistry, Chinese Academy of Sciences, Beijing 100190, China. [2] School of Chemical Sciences, University of Chinese Academy of Sciences, Beijing 100049, China. [3] Key Laboratory of Soil Environment and Pollution Remediation, Institute of Soil Science, Chinese Academy of Sciences, Nanjing 210008, China. [4] School of Physical Sciences, CAS Key Laboratory of Vacuum Physics, University of Chinese Academy of Sciences, Beijing 100049, China. [5] CAS Center for Excellence in Topological Quantum Computation, University of Chinese Academy of Sciences, Beijing 100049, China. [6] Institute of Theoretical and Computational Chemistry, Shaanxi Key Laboratory of Catalysis, School of Chemical & Environment Sciences, Shaanxi University of Technology, Hanzhong 723000, China. ✉email: yuxiaohu@snut.edu.cn; cycao@iccas.ac.cn

Chemoselective hydrogenation of nitroarenes is a key reaction in the fine chemical industry and has wide applications in the synthesis of pigments and pharmaceuticals[1–4]. Noble metal nanocatalysts (e.g., Pt, Au, and Pd) are usually used for this reaction[5–7]. However, noble metal catalysts are costly and their high activities usually come with unsatisfactory selectivity against many substituted nitroarenes[8–11]. Since Beller et al. reported highly selective traditional metal catalysts based on Co₃O₄@N/C and Fe₃O₄@N/C, it has sparked intensive research interest in this type of catalysts[1,12]. Among them, transition metal single-atom catalysts (SACs) with $M_1$-$N_x$-C (M = Fe, Co, Ni, x = 2–6) coordination configuration exhibited much better activities than their counterpart nanoparticles while maintaining high selectivity, owing to their maximum atom efficiency and particular electronic structure[13–18]. Recently, Wang et al. found that the electron density of Ni single atoms increased with the decrease of Ni-N coordination numbers (CN), and the capability of Ni single sites to dissociate H₂ was greatly enhanced, leading to higher catalytic activity in chemoselective hydrogenation of functionalized nitroarenes[19]. This result suggests that the catalytic activity of $M_1$-$N_x$-C can also be enhanced by adjusting the coordination structure of transition metal SACs.

In most SACs, the central metal atoms were stabilized by coordination bonds with N, S, O, etc. atoms within support matrix[20–29]. The electronic and geometric structures of central metal atoms can be adjusted by tailoring the coordination environment, which would change the absorption energy of reactants on metal atoms and thus influence the catalytic process[20,23,28,30]. For SACs with $M_1$-$N_x$-C sites, the symmetric electronic distribution may limit the activation of reactants, thereby leading to hampered catalytic kinetics and performances[20,31]. Recent studies have found that introducing heteroatom P for an unsymmetrical N/P mixed-coordination can further modulate the electronic properties of center metal atoms. The unsymmetrical geometric structure can evoke the distortion of electronic density and alter the d-band center[20,23,28,31,32]. For example, Yuan et al. prepared an N/P dual-coordinated Fe single-atom catalyst, which was more favorable for the adsorption of oxygen intermediates for ORR in fuel cell[23]. Li et al. reported that a $Fe_1$-$N_3P_1$ single-atom nanozyme exhibited peroxidase-like catalytic activity, and the high activity was ascribed to the less positive charge on Fe atoms as P atoms are electron donors[32]. Thus, we anticipated that constructing the unsymmetrical N/P dual-coordinated transition metal SACs would improve the catalytic performance for the hydrogenation of nitroarenes.

In this work, we report an N/P dual-coordinated Co SAC (denoted as Co₁/NPC) with $Co_1$-$N_3P_1$ coordination structure and investigate its catalytic performance for chemoselective hydrogenation of nitroarenes. The single atomic feature and coordination structure of the $Co_1$-$N_3P_1$ site are characterized through aberration-corrected high angle annular dark-field scanning transmission electron microscopy (AC HAADF-STEM), atomic-resolution electron energy-loss spectroscopy (EELS), X-ray photoelectron spectroscopy (XPS), and X-ray absorption spectrum (XAS). In nitrobenzene hydrogenation reaction, the $Co_1$-$N_3P_1$ SAC exhibits a turnover frequency of 6560 h⁻¹, which is 60 times higher than that of $Co_1$-$N_4$ SAC and 10 times higher than the state-of-the-art $M_1$-$N_x$-C SACs in literatures. Furthermore, $Co_1$-$N_3P_1$ SAC shows superior selectivity (>99%) toward many substituted nitroarenes with the co-existence of other sensitive reducible groups. The unprecedentedly high activity of $Co_1$-$N_3P_1$ SAC can be ascribed to the upshift d-band center of Co single atoms, which is more favorable for H₂ dissociation as verified by the kinetic isotope effect and density functional theory calculation results. This is an excellent example of such an unsymmetrical N/P dual-coordinated structure of metal SACs in hydrogenation.

## Results

**Structural characterization.** Supplementary Fig. 1 illustrates the synthesis procedures for preparing N/P dual-coordinated Co SAC (denoted as Co₁/NPC) via a two-step process. First, tannic acid, (2-Aminoethyl)phosphonic acid (AePA), and cobalt ion precursors were co-adsorbed on the surface of graphitic carbon nitride (g-C₃N₄) nanosheets; then the resultant powder was subjected to pyrolysis under flowing Ar gas at 900 °C to obtain Co₁/NPC, where the AePA was absent and the introduced-P species served as the donors for anchoring Co atoms (Supplementary Fig. 2). For comparison, N-coordinated Co SAC (denoted as Co₁/NC) was also prepared via the same procedure without the addition of AePA. As exhibited in Raman spectra, the carbon matrices in both Co₁/NPC and Co₁/NC were disordered with a large number of defects (Supplementary Fig. 3). Only two broad peaks at ~24.3° and 42.6° could be observed from their X-ray diffraction (XRD) patterns (Supplementary Fig. 4), corresponding to (002) and (101) planes of carbon, suggesting highly dispersed states of Co species in both of two samples. Further increasing the pyrolysis temperature of Co₁/NPC to 1000 °C led to the formation of Co₂P nanoparticles (Supplementary Figs. 4, 5). Scanning electron microscopy (SEM) and transmission electron microscopy (TEM) images show that both catalysts retain a two-dimensional layered structure and no obvious nanoparticles are observed (Supplementary Figs. 6, 7). Energy-dispersive spectroscopy (EDS) mappings reveal Co elements are distributed uniformly over the entire samples (Supplementary Figs. 8, 9). Additionally, Co single-atom feature in Co₁/NPC and Co₁/NC is directly observed by AC HAADF-STEM, as reflected by the highly dispersed bright dots due to the heavy Z-contrast (Fig. 1a and Supplementary Fig. 10). The Co loading in Co₁/NPC was ~0.45 wt% as determined by the inductively coupled plasma mass spectroscopy (ICP-MS) analysis (Supplementary Table 1). The porosity features of Co₁/NC and Co₁/NPC were investigated using nitrogen physisorption measurements. Both catalysts displayed characteristics of IV-type N₂ adsorption-desorption isotherms, suggesting the existence of mesopores, which would be beneficial for the exposure of active sites and mass transportation (Supplementary Fig. 11). The calculated BET-specific surface area of Co₁/NC and Co₁/NPC were 512 and 471 m² g⁻¹, respectively.

XPS was then applied to reveal the chemical structures of both Co SACs. In N 1s spectra, besides pyridinic N, pyrrolic N, graphitic N, and oxidized N species, a peak at 399.1 eV corresponding to Co-N can be distinguished[33], indicating the existence of N coordination environment with Co single atoms in both catalysts (Fig. 1b and Supplementary Fig. 12). Note that from P 2p spectra in Fig. 1c, an obvious peak at ~129.3 eV corresponding to Co-P bond was presented in Co₁/NPC[23,31], which can also be observed in the comparison Co₂P NPs/C (Supplementary Fig. 13); while it is absent in Co₁/NC sample. These results suggest that the atomically dispersed Co atoms possess N/P dual-coordinated configuration in Co₁/NPC, while only N-coordinated configuration in Co₁/NC.

To further determine the coordination environment of Co single atoms, X-ray absorption fine structure (XAFS) measurements were conducted. Figure 1d shows the Co K-edge X-ray absorption near-edge structure (XANES) curves of Co₁/NC and Co₁/NPC, with Co foil, CoO, and cobalt phthalocyanine (CoPc) as reference samples. It can be seen that the absorption threshold positions for Co₁/NC and Co₁/NPC are located between Co foil and CoO, suggesting that the valence states of Co species are between 0 and +2 in both two catalysts. Moreover, the Co K-edge position and white line of Co₁/NPC are lower than that of Co₁/NC (inset of Fig. 1d), which indicates that Co atoms in Co₁/NPC possess more negative charges than Co₁/NC. Such difference could be attributed to the less electron transfer from Co to P

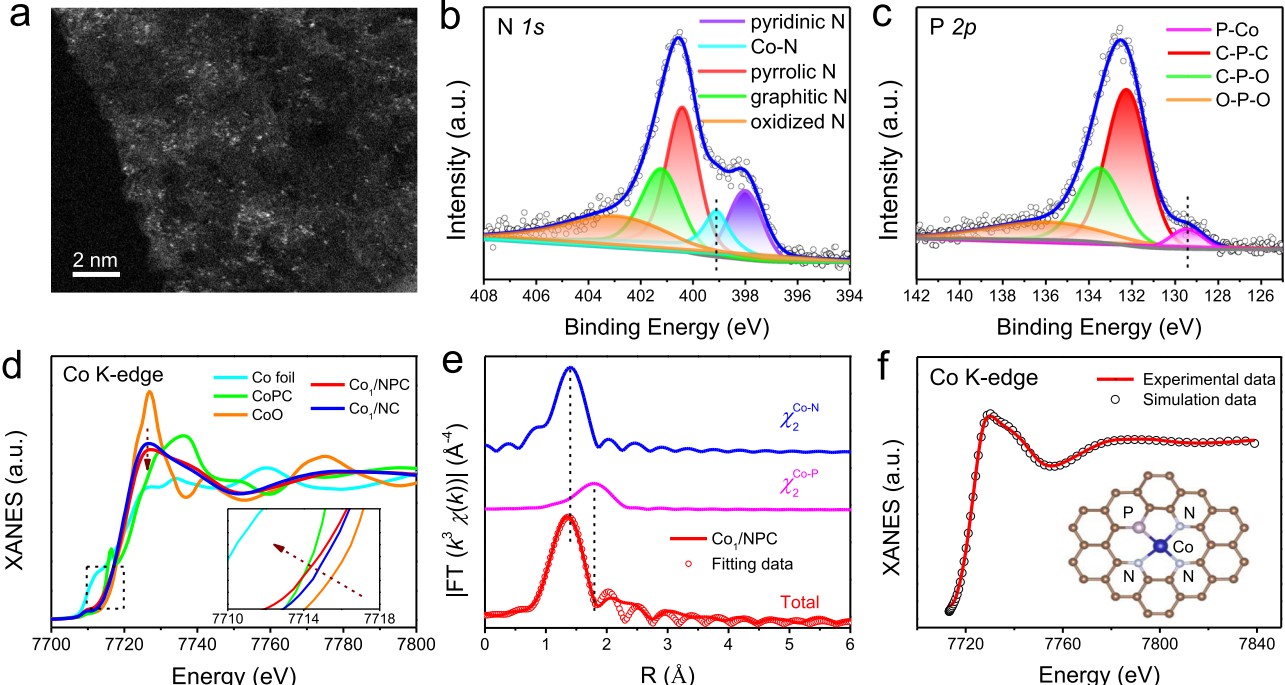

**Fig. 1 Structural Characterizations of Co₁/NPC. a** AC HAADF-STEM image of Co₁/NPC. **b** N 1s and **c** P 2p XPS spectra of Co₁/NPC. **d** Co K-edge XANES spectra. **e** $k^3$-weight FT-EXAFS fitting curves of Co₁/NPC. Curves from top to bottom are the Co-N, Co-P backscattering pathways and fitting total signal (red circle) superimposed on the experimental signal (red line). **f** The experimental XANES curve in comparison with the calculated XANES data of $Co_1$-$N_3P_1$ site in Co₁/NPC sample. Inset: the schematic atomic structure of Co₁/NPC derived from the EXAFS results.

because of the weaker electronegativity of P than N atoms[20]. The Fourier-transformed $k^3$-weighted EXAFS (FT-EXAFS) spectra demonstrated that both Co₁/NC and Co₁/NPC only exhibited a prominent peak at 1.38 Å (without phase shift), no Co-Co peaks at 2.17 Å or larger bond distances were detected, confirming atomically dispersed Co species in Co₁/NC and Co₁/NPC (Supplementary Figs. 14, 15). The coordination configuration of Co moieties was further surveyed using quantitative least-squares EXAFS curve-fitting. The EXAFS spectrum of Co₁/NPC was investigated by utilizing Co-N and Co-P backscattering pathways. The best-fitting analysis displays that the main peak at 1.38 Å could be satisfactorily interpreted as Co-N first-shell coordination with $CN = 3.2 \pm 0.1$ and the shoulder peak at 1.77 Å originated from Co-P contribution with $CN = 0.9 \pm 0.1$ (Fig. 1e and Supplementary Table 2), suggesting the possible $Co_1$-$N_3P_1$ configuration in Co₁/NPC. For comparison, fitting of Co₁/NC resulted in an average of about four N atoms with a distance of 1.37 Å (Supplementary Fig. 14 and Supplementary Table 2). In order to better confirm the proposed configurations, the theoretical XANES spectrum were simulated based on the $Co_1$-$N_3P_1$ model as well as $Co_1$-$N_4$, which presented a good agreement with the experimental data, indicating the rationality of the two structures (Fig. 1f and Supplementary Fig. 16).

Besides, we performed atomic-resolution electron energy-loss spectroscopy (EELS) analysis at a relatively low beam current to minimize electron-beam perturbations to provide strong evidence of $Co_1$-$N_3P_1$ structure (Fig. 2a, b). The extracted Co L-edge EELS spectrum from Fig. 2d presents a clear Co signal (Fig. 2g), providing direct evidence for the presence of atomically dispersed Co species. More importantly, the existence of N, P dual-coordination vicinal to Co site is revealed by identifying the surrounding heteroatoms. From the N, P, and overlap maps (Fig. 2e, f, c), three N atoms (green) and one P atom (red) exist around the Co site. Atomic-scale N K-edge and P L-edge EELS spectra collected at the corresponding positions from Fig. 2e, f are

further demonstrated by the N and P signals (Fig. 2h, i). This forcefully confirms the $Co_1$-$N_3P_1$ configuration in Co₁/NPC sample. Moreover, the formation energy of the $Co_1$-$N_3P_1$ structure in the Co₁/NPC sample was estimated to be about −0.864 eV by DFT calculation, indicating the high stability of the proposed configuration (Supplementary Fig. 17). These results revealed that the Co single sites in Co₁/NPC were stabilized with N/P dual-coordinated structure, forming an unsymmetrical $Co_1$-$N_3P_1$ geometric configuration (as depicted in Fig. 1f), which is different from the Co site in Co₁/NC with traditional in-plane $Co_1$-$N_4$ configuration.

**Catalytic performance.** To evaluate the catalytic performances of the as-prepared Co SACs for the hydrogenation of nitroarenes, nitrobenzene is first chosen as a probe molecule. The reaction kinetics with Co₁/NC and Co₁/NPC were obtained at 110 °C with 3 MPa $H_2$ in a Teflon-lined stainless steel autoclave. As shown in Fig. 3a, Co₁/NPC exhibits significantly higher activity than that of Co₁/NC. Nitrobenzene was completely converted with >99% amine selectivity in 210 min with Co₁/NPC, while less than 20% conversion was observed with Co₁/NC under the same reaction condition. In addition, no conversion was observed with NC and NPC supports, suggesting that atomic Co site was active species in both Co₁/NC and Co₁/NPC catalysts (Supplementary Fig. 18). The reaction rate (k) for hydrogenation of nitrobenzene over Co₁/NPC could reach as high as 35.9 mol mol⁻¹ min⁻¹, which is ten times higher than that with Co₁/NC (3.1 mol mol⁻¹ min⁻¹).

The turnover frequency value (TOF) (based on the substrate conversion at about 20%) of Co₁/NPC is calculated to be 6560 h⁻¹, which is over 60 times higher than that with Co₁/NC (108 h⁻¹) (Fig. 3b). Besides nitrobenzene, Co₁/NPC also exhibited superior high activity and excellent selectivity (>99.7%) for hydrogenation of 3-nitrostyrene with TOF of 4499 h⁻¹. The impressive activity of Co₁/NPC is ten times higher than the state-of-the-art $M_1$-$N_x$-C SACs in

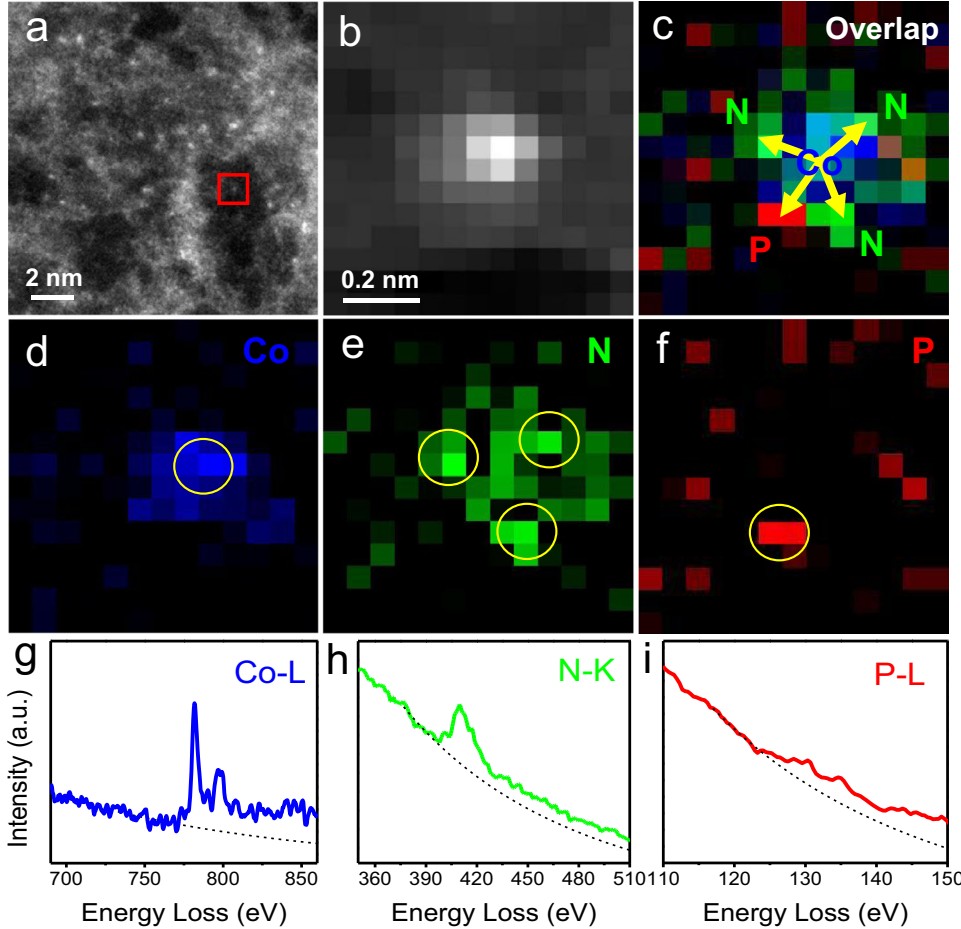

**Fig. 2 Atomic-resolution EELS analysis of $Co_1$/NPC sample. a** AC HAADF-STEM image of $Co_1$/NPC sample at a relatively low beam current. **b** HAADF image acquired simultaneously with atomic-resolution EELS mapping of $Co_1$/NPC at the red square area in (**a**). **c** The overlap map of P, N, and Co elements. **d–f** Elemental maps of Co, N, P, respectively. **g–i** The EELS spectra extracted at the yellow circled position from (**d–f**), respectively.

literature (Fig. 3c and Supplementary Table 3)[34–43]. Such excellent catalytic performance of $Co_1$/NPC sample inspired us to carry out the reaction under milder conditions (e.g., 40 °C, 1 bar $H_2$). A high nitrobenzene conversion of 97.2% was achieved within 5 h (Supplementary Fig. 19). To further compare the catalytic performance between $Co_1$/NC and $Co_1$/NPC, the apparent activation energies of these two catalysts were measured (Supplementary Fig. 20 and Supplementary Table 4). As shown in Fig. 3d, the calculated activation energy of $Co_1$/NPC catalyst is about 21.8 kJ mol$^{-1}$, which is much lower than that of $Co_1$/NC (51.3 kJ mol$^{-1}$).

In order to clarify the intrinsic higher activity of $Co_1$-$N_3P_1$, the electronic properties of the central metal sites over $Co_1$/NC and $Co_1$/NPC are examined by electron-density isosurface and partial density of states from DFT calculations. Different charge distributions of the two models are observed (Fig. 3e). Compare to the $Co_1$-$N_4$ configuration, the symmetric electron structure is broken by introducing heteroatom P in $Co_1$-$N_3P_1$. The Bader charge of the $Co_1$-$N_3P_1$ site is estimated to be +0.81 $e$, while the $Co_1$-$N_4$ site is +0.97 $e$, indicating the Co atom in the $Co_1$-$N_3P_1$ site carries more charge since the P element in the $Co_1$/NPC transfers 2.308 $e$ to support, which is consistent with the XAFS results. Moreover, the Co d-band center of $Co_1$/NPC is up-shifted, much closer to the Fermi level (Fig. 3f). As a result, the antibonding state of Co atoms and adsorbed $H_2$ species are more occupied, then such change enhances the capabilities of $H_2$ dissociation[6,44]. Thus, $Co_1$/NPC catalyst with $Co_1$-$N_3P_1$ configuration exhibits much higher activity than that of $Co_1$/NC with $Co_1$-$N_4$ configuration.

**Catalytic hydrogenation mechanism.** Such a large activity difference between $Co_1$/NC and $Co_1$/NPC implies that the hydrogenation activity is closely correlated with local coordination structure, which influences the electronic structure of Co single atoms. In order to elucidate the reaction mechanisms on both catalysts, we carried out a kinetic isotope effect (KIE) study to examine the $H_2$ dissociation step. Using $D_2$ for nitrobenzene hydrogenation, the reaction rate was slowed down by a factor of 3.25 for $Co_1$/NC (Fig. 4a). For comparison, a larger KIE ($k_H/k_D = 5.54$) was observed on $Co_1$/NPC catalyst (Fig. 4b). These results suggest that $H_2$ dissociation undergoes heterolytic cleavage on both $Co_1$/NC and $Co_1$/NPC[45–47].

It is generally accepted that heterolytic cleavage of $H_2$ on metal single atoms occurs to form metal-$H^{\delta-}$ and heteroatom-$H^{\delta+}$ [45,48]. Therefore, DFT calculations were further performed to understand the $H_2$ heterolytic cleavage on $Co_1$/NC and $Co_1$/NPC, respectively. As shown in Supplementary Fig. 21, the $H_2$ molecule is adsorbed on the Co atom of $Co_1$/NPC with adsorption energy of −0.08 eV, and the bond length of $H_2$ is 0.814 Å, which is much longer than free molecular $H_2$ (0.752 Å) in the gas phase. While $H_2$ molecule is adsorbed on Co atom of $Co_1$/NC with adsorption energy of 0.033 eV, and the bond length of $H_2$ is 0.783 Å. The much longer H-H bond length of adsorbed molecule $H_2$ indicates that $Co_1$/NPC has a higher activation ability for $H_2$ dissociation than $Co_1$/NC. Subsequently, one of the H atoms moves to a nearby heteroatom (N and P) to yield heteroatom-$H^{\delta+}$, leaving another H atom on Co atom as Co-$H^{\delta-}$. Direct dissociation of $H_2$

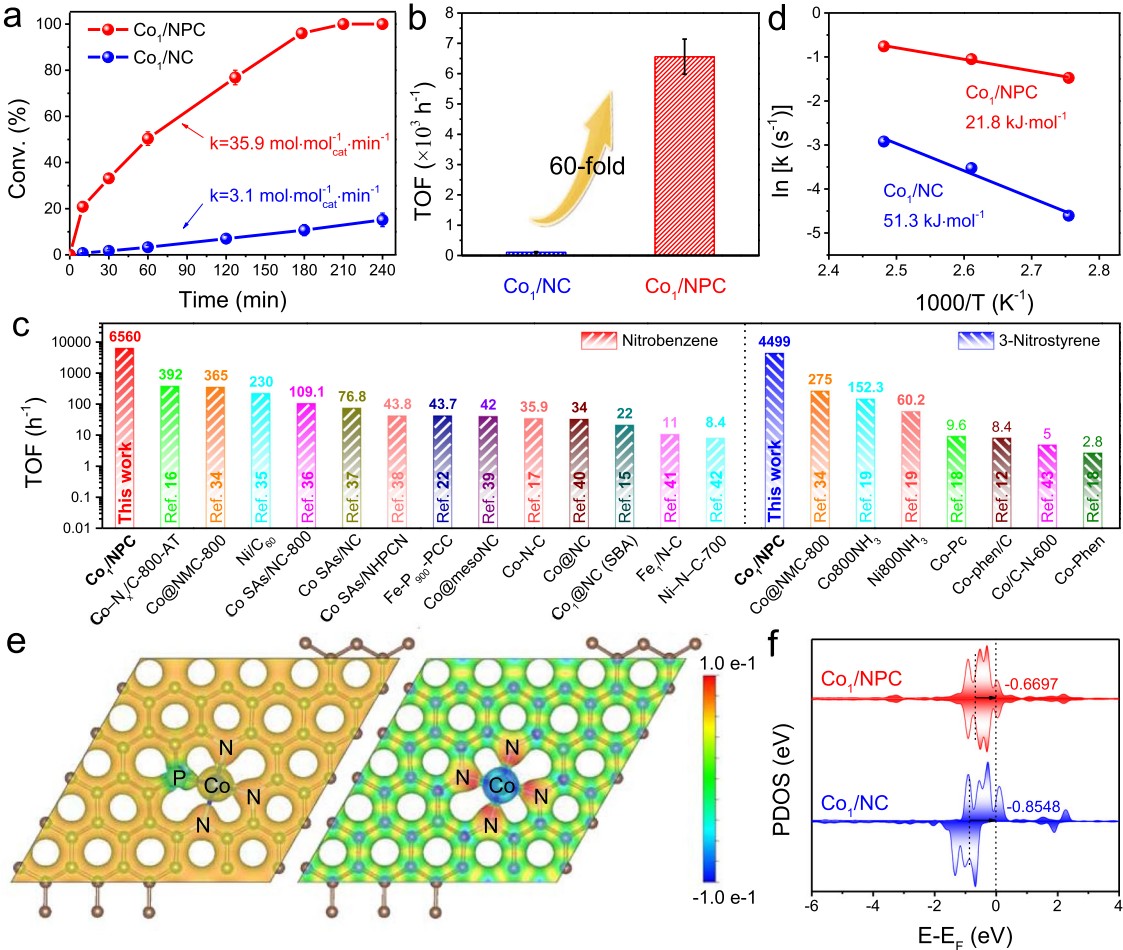

**Fig. 3 Catalytic performances and kinetic process studies. a** Time course of nitrobenzene conversions over $Co_1$/NPC and $Co_1$/NC samples. The standard deviations were derived from three independent trials. Reaction conditions: 5 mg catalyst, 2 mmol nitrobenzene, 40 mL $EtOH/H_2O$ (v:v = 4:1), 110 °C, 3 MPa $H_2$. **b** TOF of $Co_1$/NPC and $Co_1$/NC for the nitrobenzene hydrogenation. The TOF values were calculated at about 20% conversion. **c** The TOF values comparison of recently reported transition metal catalysts. **d** The experimental Arrhenius plots of $Co_1$/NPC and $Co_1$/NC. **e** Electron-density isosurface of Co atoms in two models. Blue color indicates positive charges and red color indicates negative charge. **f** Partial density of states (PDOS) of Co atoms in $Co_1$/NPC and $Co_1$/NC, the zero-energy corresponds to the Fermi level, and the d-band centers are inserted with the short dot.

on both $Co_1$/NPC and $Co_1$/NC in the absence of water, the transition state is almost the same (~1.21 eV), the only difference is that the dissociation of $H_2$ on $Co_1$/NPC is exothermic by 0.20 eV, while on $Co_1$/NPC is endothermic by 1.11 eV, suggesting such dissociation manner on $Co_1$-$N_4$ site is thermodynamically unfavorable (Supplementary Fig. 22). The above difference in the DFT calculations confirms that $Co_1$-$N_3P_1$ exhibits much higher catalytic activity for heterolytic cleavage of $H_2$.

Ding et al.[18]. reported that the protic solvents play a dominant role in the case of Co-N-C-catalyzed hydrogenation of nitroarenes, where the solvent-mediated H-shuttling mechanism is crucial in the reaction pathway. Compared to the intrinsic hydrogen transfer, the protic solvent-mediated one usually possesses a lower activation barrier, leading to an enhancement of hydrogenation activity in the presence of water or alcohol[49]. Indeed, both $Co_1$/NC and $Co_1$/NPC show the best activities under ethanol/water solvent and significantly decreased activities in an aprotic solvent such as toluene, acetonitrile, THF, and n-hexane (Supplementary Fig. 23). Further DFT calculations suggest that the activation energy barriers with water-mediated H-shuttling mechanism for the heterolytic cleavage of $H_2$ are lower by about 0.16 and 0.01 eV than that through direct dissociation on $Co_1$-$N_3P_1$ and $Co_1$-$N_4$ sites, respectively (Fig. 4c, d). Both kinetic and

thermodynamic results suggest that the dissociative activation of $H_2$ with help of $H_2O$ is more favorable to $Co_1$/NPC catalyst.

According to the previous reports, the hydrogenation reduction of nitrobenzene to aniline follows the Haber mechanism[7], namely, $PhNO_2^* → PhNOOH^* → PhNO^* → PhNOH^* → PhNHOH^* → PhNH^* → PhNH_2^*$. Based on the above results and reported mechanism in literature, the reaction pathway for hydrogenation of nitrobenzene over $Co_1$-$N_3P_1$ catalyst is further proposed by virtue of DFT calculations, as shown in Fig. 5 and Supplementary Fig. 24. One $H_2$ molecule first goes through heterolytic cleavage with the assistance of the $H_2O$-mediated H-shuttling mechanism to form Co-$H^{\delta-}$ and P-$H^{\delta+}$ at $Co_1$-$N_3P_1$ sites, which can serve as the initial state for the hydrogenation process (Fig. 5a and Supplementary Fig. 25). Then, the target nitrobenzene molecule was adsorbed on the $Co_1$-$N_3P_1$ site with a free energy of −0.78 eV (Fig. 5b, I). Subsequently, the activated H atom on Co-$H^{\delta-}$ and the O atom of $PhNO_2$ are combined to produce PhNOOH intermediate, which is later reduced to PhNO intermediate by the H atom transfer from P-$H^{\delta+}$ (II). Notably, the PhNO intermediate can be detected during the reaction process (Supplementary Fig. 26). After that, another $H_2$ molecule is dissociated to form an activated H atom, which attacks the oxygen atom of PhNO and reduces it to PhNHOH (III, IV). It is worth

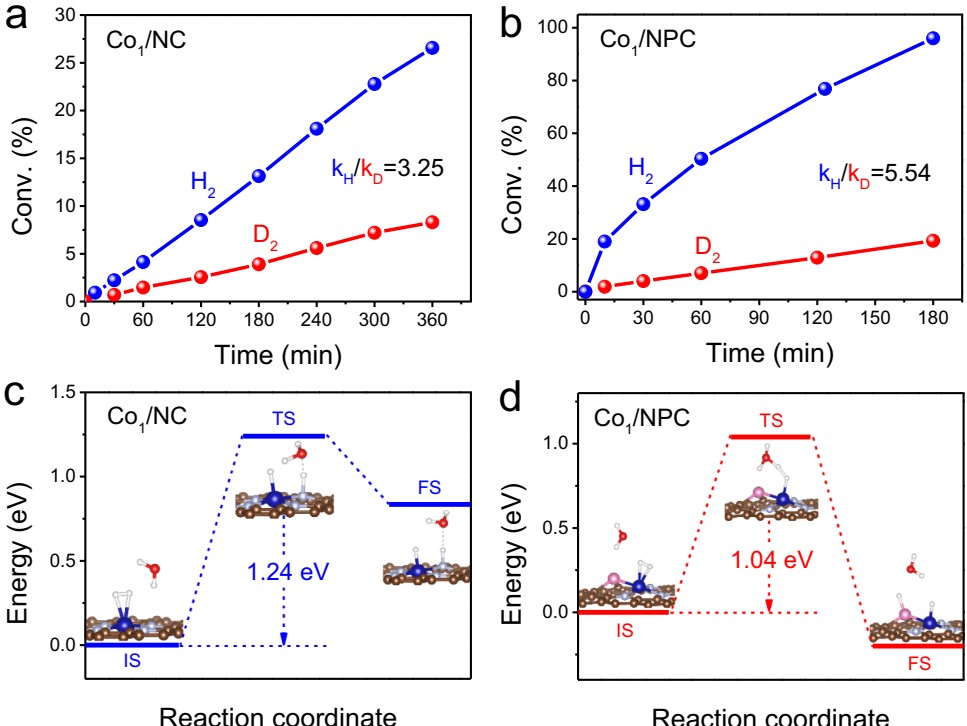

**Fig. 4 Catalytic mechanism for the hydrogenation of nitrobenzene.** Primary isotope effect observed on **a** $Co_1$/NC and **b** $Co_1$/NPC. Reaction conditions: 5 mg catalyst, 2 mmol nitrobenzene, 40 mL EtOH/$H_2O$ ($D_2O$) (v:v = 4:1), 110 °C, 3 MPa $H_2$ ($D_2$). Energies profiles for $H_2$ dissociation pathways of **c** $Co_1$/NC and **d** $Co_1$/NPC samples. IS initial state, TS transition state, FS final state.

noting that the adsorption energies of intermediates PhNO and PhNHOH on the $Co_1$-$N_3P_1$ site are more favorable than Ph-$NO_2$, ensuring the reaction progress of the targeted substrate (Supplementary Fig. 27). In the next step, the third $H_2$ molecule participates in and the formed H atom interacts with PhNHOH to generate the final $PhNH_2$ product (V, FS). It can be seen the whole process is highly endothermic, confirming the possibility of the proposed reaction path.

**Substrate exploration and catalytic stability.** A broad scope of substituted nitroarenes was tested to examine the chemoselectivity in nitroarene hydrogenation (Fig. 6). $Co_1$/NPC shows impressive chemoselectivity toward the substituted nitroarenes in the presence of other sensitive reducible groups, such as alkenyl (99.7%, 2b), halogen (>98.9%, 2c–f), ketones (>99%, 2g, h), nitrile groups (>99%, 2i), etc. Notably, $Co_1$/NPC also exhibits high activity and selectivity toward heterocyclic nitro-compounds (>99%, 2n–q). The superior high selectivity to corresponding anilines is ascribed to the unique character of metal SACs, where there is only one single metal atom for adsorption and activation of substrates.

Furthermore, $Co_1$/NPC exhibited tolerable stability. As shown in Supplementary Fig. 28, a slight decrease of activity is observed after five cycles with $Co_1$/NPC, which can be ascribed to the loss of some catalysts and active Co species during the recycling experiments (Supplementary Table 1). The AC HAADF-STEM image and Co K-edge EXAFS spectrum of spent $Co_1$/NPC indicate that the atomically dispersed Co species is well preserved after five cycles (Supplementary Fig. 29). All these results demonstrate that the $Co_1$/NPC catalyst with unsymmetrical $Co_1$-$N_3P_1$ configuration possesses unprecedented high activity, high selectivity, and good stability to a wide scope of substrates for hydrogenation of nitroarenes.

## Discussion

In summary, we produced an atomically dispersed $Co_1$/NPC catalyst with an unsymmetrically $Co_1$-$N_3P_1$ coordination structure. Due to the increased electron density and upshift d-band center of Co atoms in $Co_1$-$N_3P_1$, $H_2$ dissociation was proved to be more favorable, resulting in much enhanced catalytic activity. In nitrobenzene hydrogenation reaction, the as-prepared $Co_1$-$N_3P_1$ SAC exhibited a 60-fold higher TOF value (6560 h$^{-1}$) than that of $Co_1$-$N_4$ SAC and more than tenfold higher than the state-of-the-art $M_1$-$N_x$-C SACs in literature. In addition, $Co_1$-$N_3P_1$ SAC also displayed superior selectivity (>99%) towards the substituted nitroarenes with the co-existence of other sensitive reducible groups. This work provides new insight into rationally modulating the coordination structure of central metal atoms for boosting the catalytic performance of SACs in heterogeneous catalysis.

## Methods

**Synthesis of $Co_1$/NPC and $Co_1$/NC.** In a typical procedure, $Co(NO_3)_2 \cdot 6H_2O$ (6.5 mg), tannic acid (TA, 500 mg), and (2-Aminoethyl) phosphonic acid (AePA, 126 mg) were dissolved into 30 mL DI water at 100 °C (marked as solution A). g-$C_3N_4$ nanosheets (1 g) were dispersed well in 100 mL DI water with ultrasound (marked as solution B). Then, solution A was added dropwise into solution B with a strong stirring at 100 °C until the mixed system was forced to yield a slurry. Subsequently, the obtained powder after freeze-dried was pyrolyzed at 900 °C for 2 h under Ar atmosphere. Finally, the as-prepared material was directly used without further treatment, denoted as $Co_1$/NPC.

The synthesis process for $Co_1$/NC is the same as that of $Co_1$/NPC except without the addition AePA.

**Characterizations.** The powder X-ray diffraction (XRD) patterns were recorded on a Rigaku D/max-2500n diffractometer with Cu Kα radiation (λ = 1.5418 Å) at 40 kV and 200 mA. The morphologies and microstructures of the samples were measured on the transmission electron microscopy (TEM) (JEM-2100F, JEOL, Japan) and the scanning electron microscopy (SEM) (HITACHI S-4800, Japan). Element mapping was characterized on TEM equipped with Oxford detection. X-ray photoelectron spectroscopy (XPS) measurements were performed on a VG

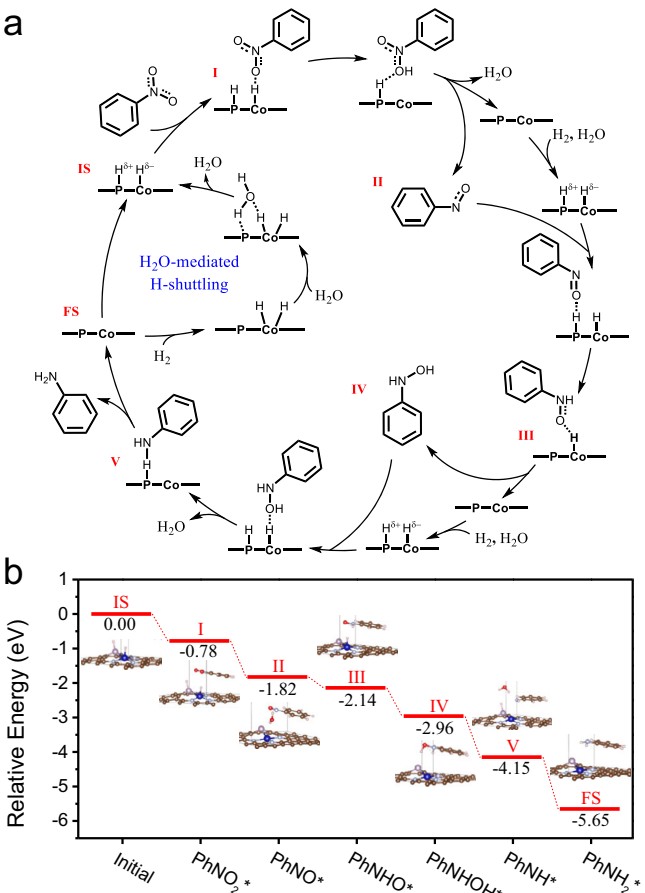

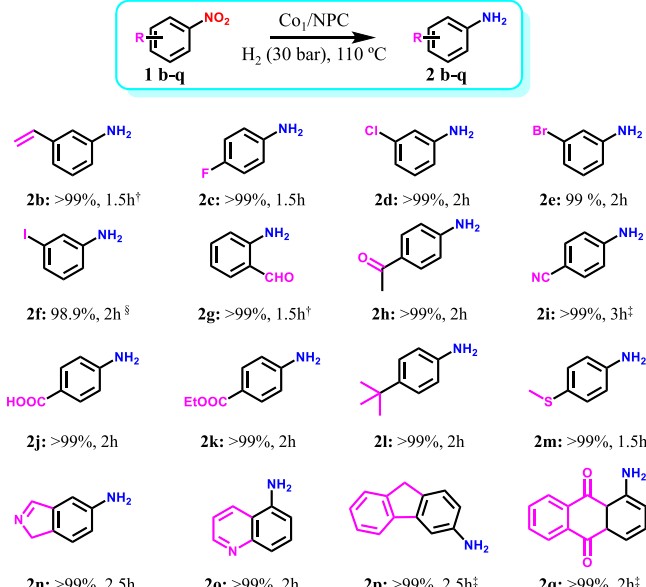

**Fig. 5 Reaction mechanism of the nitrobenzene reduction. a** The proposed reaction pathway for the hydrogenation of nitrobenzene to aniline at Co-P interface site. **b** Energy profile of hydrogenation of nitrobenzene over Co$_1$-N$_3$P$_1$ site.

**Fig. 6 Substrate scope of hydrogenation over the Co$_1$/NPC catalyst.**
Reaction conditions: 1 mmol nitroarenes; 10 mL EtOH; 5 mg catalyst; 110 °C; 3 MPa H$_2$. †EtOH/H$_2$O, v-v = 4:1, 10 mL; §H$_2$, 2 MPa; ‡temperature, 120 °C. In all cases, complete conversions of nitroarenes were observed.

Scientific ESCALab220i-XL electron spectrometer using 300 W Al kα radiation. Inductively coupled plasma atomic emission spectroscopy (ICP-AES) was conducted on a Shimadzu ICPE-9000 to confirm the loading content of metal on the catalysts. The AC HAADF-STEM images were carried out in a JEOL ARM300F at 300 kV, equipped with a probe spherical aberration corrector.

Brunauer–Emmett–Teller (BET) surface areas were measured by N$_2$ adsorption-desorption isotherms at 77 K with a Micromeritics ASAP 2460 instrument. The HAADF imaging and EELS mapping were both performed using a Nikon HERMES-100 aberration-corrected scanning transmission electron microscope under 60 kV accelerating voltage with a ~22 pA probe. The probe convergence semi-angle, HAADF collection semi-angle, and EELS collection semi-angle is 32 mrad, 75–210 mrad, and 0–75 mrad, respectively.

**XAS measurements and analysis**. The cobalt K-edges XAFS spectra of the standards and samples were collected at the beamline 1W1B of the Beijing Synchrotron Radiation Facility (BSRF). The typical energy of the storage ring was 2.5 GeV and the electron current was ~250 mA in the top-up mode. The white light was monochromatized by a Si (111) double-crystal monochromator and calibrated with a Co foil (K-edge at 7709 eV). Samples were pressed into thin slices and positioned at 45° to the incident beam in the sample holder. The XAFS spectra were recorded in fluorescence mode with a Lytle detector oriented at 90° to the incoming beam.

The XAFS data were analyzed using the software packages Demeter[50]. The spectra were normalized using Athena firstly, and then shell fittings were performed with Artemis. The χ(k) function was Fourier-transformed (FT) using $k^3$ weighting, and all fittings were done in R-space. The coordination parameters of samples were obtained by fitting the experimental peaks with theoretical amplitude. The quantitative curve-fittings were conducted with a Fourier transform k-space range of 2.7–11.8 Å$^{-1}$. The backscattering amplitude $F$(k) and phase shift Φ(k) were calculated by the FEFF7.0 code. While the curve-fitting, all the amplitude reduction factor S$_0^2$ was set to the best-fit value of 0.87 determined from fitting the data of cobalt foil by fixing coordination numbers as the known crystallographic value. In order to fit the curves in the R-range of 1.0–2.0 Å, we considered Co-N and Co-P paths as the central-peripheral. For each path, the structural parameters, like coordination number (CN), interatomic distance (R), Debye–Waller factors (σ$^2$), and inner potential correction (ΔE$_0$) were opened to be varied.

**Catalytic performance evaluation**. For the liquid phase hydrogenation of nitrobenzene, substrates (2 mmol), catalyst (5 mg), and solvent (EtOH/H$_2$O, v-v = 4:1, 40 mL) were added into a 100 mL high-pressure autoclave. Then the autoclave was flushed three times with H$_2$ and charged to certain pressure (3 MPa H$_2$). The reaction was performed at the desired temperature. The product was collected at the reserved time and immediately analyzed using gas chromatography in combination with mass spectrometry (Shimadzu GCMS-QP2010S). The TOF values were determined within the substrate conversion below 20%, and the calculation of TOF was based on the total Co amount in catalysts. To evaluate the reusability of catalyst, the samples from the last reaction were separated by centrifugation, washing with ethyl acetate to remove the substrate, and drying under the vacuum. For the kinetic isotope effect test, D$_2$ and D$_2$O are used instead of H$_2$ and H$_2$O.

**DFT calculations**. All the spin-polarized first-principles calculations have used the code VASP[51–53]. Valence electrons of O(2s, 2p), N(2s, 2p), H(1s), C(2s, 2p), P(3s, 3p), and Co(3d, 4s) were treated on a basis of plane waves explicitly[54], while the core electrons were described with the projector-augmented wave method[55]. Spin-polarized calculations were carried out at the level of the generalized gradient approximation (GGA) adopting the Perdew, Burke, and Ernzerhof (PBE) exchange-correlation functional[56]. A kinetic energy cutoff of 400 eV was used for all calculations. The truncation criteria for the electronic and ionic loops were 10$^{-5}$ eV and 10$^{-2}$ eV/Å, respectively. Long-range dispersion was included according to the D$_3$ method introduced by Grimme[54]. The vacuum layer was set to 20 Å to avoid interaction from adjacent cells. All the transition states were determined by using the climbing image nudged elastic band (CINEB) method[57,58] and transition states were characterized via frequency analysis to ensure a single imaginary frequency in the desired reaction direction. The pure graphene is modeled by a (7 × 7) supercell with 49 carbon atoms, and the Co-N$_x$P$_y$-Gr model is modeled by one cobalt atom adsorption at vacancy site which is composed by getting rid of two carbon atoms and the N and P atoms substitute for four carbon atoms around cobalt atom, respectively. Monkhorst–Pack (5 × 5 × 1) Γ-centered grid sampling for the Brillouin zone was used for geometry optimization, and dipole corrections were included in the z-direction for each model surface. To study the stability of Co binding on graphene with N, P dual-coordination, the formation energy, and adsorption energy were defined as follows:

$$E_{for} = E(Co-N_xP_y-Gr) - E(N_xP_y) - E(Co-bulk) \quad (1)$$

$$E_{ads} = E(Co-N_xP_y-Gr) - E(N_xP_y) - E(Co) \quad (2)$$

In the equations, E(Co-N$_x$P$_y$-Gr) is the total energy of adsorbed systems, E(N$_x$P$_y$) is the energy of graphene with doped N and P, E(Co-bulk) is the energy of one atom in the most stable Co crystal, and E(Co) is the energy of cobalt in the gas phase.

## Data availability

The data supporting the findings of this study are available within the article and its Supplementary Information. Source data are provided with this paper. Additional data are available from the corresponding authors on reasonable request.

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

## Acknowledgements

We thank the National Key R&D Program of China (Grant No. 2018YFA0703503 and 2018YFA0208504, C.C. and W.S.), the National Natural Science Foundation of China (NSFC 21932006, W.S.), the Youth Innovation Promotion Association of CAS (2017049, C.C.), Beijing Outstanding Young Scientist Program (BJJWZYJH01201914430039, W.Z.) and National Science Basic Research Program of Shaanxi (No. S2020-JC-WT-0001, X.Y.) for financial support. We thank the beamline 1W1B station in Beijing Synchrotron Radiation Facility (BSRF) and Dr. Lirong Zheng for help in XAFS characterization.

## Author contributions

H.J., C.C., and W.S. were responsible for most of the investigations, methodology development, data collection/analysis, and writing the original manuscript. P.L. assisted with the experiments analysis. X.Y. conducted the DFT calculations. P.C. helped to analyze the XAFS results. J.S. and W.Z. helped to conduct the atomic-resolution EELS. C.C. and W.S. were responsible for the funding and resources acquisition, supervising the project, revising, and editing the manuscript.

## Competing interests

The authors declare no competing interests.
