## [Peer Review File · Nature Communications]

Title: Unprecedented high activity and chemoselective for hydrogenation of nitroarenes with single atomic Co₁-N₃P₁ sitesREVIEWER COMMENTS

Reviewer #1 (Remarks to the Author):

The authors reported in this work a single-atom catalyst with CO₁-N₃P₁ sites that exhibits very high activity and selectivity for hydrogenation of nitroarenes. The experiments were nicely designed and very well done. The computational part, however, is not convincing. The following issues need to be addressed before I can recommend accepting the manuscript.

- 1) The reaction path shown in Fig. 4 is only a sketch that is based on the previously proposed mechanism of homogeneous catalysis. It has been well known that the heterogeneous catalysis can be very different from the homogeneous one. It's not to just 'borrow' the mechanism of homogeneous catalysis here without any support from either experiments or calculations. This is the biggest problem of the paper.
- 2) Increase in electron density of Co is a selling point of the paper. The Bader charge shall be provided to support this.
- 3) It was claimed in the paper that H₂ heterolytic cleavage will form metal-H δ^- and heteroatom-H δ^+ , but the Bader charges were not provided. Adsorption energy of individual H atoms after cleave can also be provided which will be helpful in explaining the proposed mechanism.

Reviewer #2 (Remarks to the Author):

In this work, Song et al. reported an unsymmetrically coordinated single atomic catalyst with Co₁-N₃P₁ sites for hydrogenation of functionalized nitroarenes. This is an excellent example of enhancing catalytic activity by modifying the coordination structure of single atom catalysts. The catalyst in this work shows impressive activity and selectivity in nitrobenzene hydrogenation reaction. The mechanism of the catalytic reaction was also thoroughly investigated by kinetic isotope effect and density functional theory calculations. In my opinion, this is an excellent research work and it can be published in Nature Communications after minor revision.

These issues need to be addressed before publication:

1. Freeze-drying is an important step in the preparation process of Co₁/NPC catalyst and is suggested to be involved in Scheme S1. In addition, there is a typo in line 266, where "before" should be "after".
2. It is difficult to distinguish the positions of absorption edges in Fig. 1d. It is suggested to enlarge the details or add a first derivative diagram in the supplementary information to show the differences.
3. The scale bar of Supplementary Fig. 3 is not clearly visible.
4. Other characterizations about N, P dual-coordination should be provided to further support the single site structure, such as EELS.
5. According to the proposed reaction mechanism in Figure 4, the adsorption energy of intermediates Ph-NO and Ph-NHOH on Co-H δ^- should be more favorable than Ph-NO₂. DFT calculations are suggested

to confirm this.

6. During the kinetic isotope effect study, because H₂O is also used as co-solvents and H/D exchange between D₂ and H₂O is facile. Thus, D₂ and D₂O should be noted specially in the manuscript.

7. I notice there is a little activity decline in the re-used experiments in supplementary Fig. 20. The authors should comment on the reasons for activity loss.

Reviewer #3 (Remarks to the Author):

Song and co-authors report the synthesis of Co single atom catalyst with unsymmetrical Co₁-N₃P₁ active site that displays high activity and chemoselectivity for hydrogenation of nitroarenes. Attractive catalytic performance is attributed to the increased electron density of P-modulated Co₁-N₃P₁ active site, which facilitate H₂ dissociation as verified by kinetic isotope effect and density functional theory calculations. These interesting findings make an important contribution to the field of single-atom catalysis. Before the publication, addressing following issues can further solidify this work.

1. The authors need to better explain why only one N atom has been replaced other than two or more?
2. The authors comment on whether the introduction of phosphonic acid affects the surface density of active sites of CoN₃P as compared to CoN₄.
3. How is the amplitude reduction factor S₀² determined? There are different descriptions in main text (0.712) and SI (0.87/0.90). EXAFS data in K space need to be given.
4. The authors need to better present the curves in Figure 1.d to show edge positions.
5. Please explain why there is no shoulder feature around 7717 eV for the XANES of sample Co₁/NC containing Co-N₄ configuration.
6. Please explain why the white line of XANES of sample Co₁/N₃P is sharper than that of Co₁/NC. Introducing Co-P bonding could result in the red shift of the features in XANES?
7. As the authors claim the importance of the local structure, it is highly recommended to supply the theoretical XANES simulation to further prove the proposed CoN₃P and CoN₄ structure.
8. Keep abbreviations always consistent, for example 'Fourier transformed EXAFS' in line 98 and 'FT-EXAFS' in line 99.
9. In DFT calculations part, the representation of atomic orbital should be unified.
10. The authors should provide more details about the calculated model and parameters, such as lattice parameter of model, KPOINTS, the formula of binding energy, the frequency of IS (TS and FS).
11. A number of relevant references should be cited, including Co SACs on graphene for hydrogenation of nitroarenes, tunable local structures of Co SACs and review papers related to local atomic environments of single atoms catalysis etc.

Reviewer #4 (Remarks to the Author):

This paper reports the preparation of cobalt single atom catalysts and their application for the hydrogenation of nitro compounds to anilines. In recent years, the development of single atom catalysts increases the interests of researchers. In fact, the search of new catalytic materials those should perform challenging reactions in more efficient and selective way, for those the traditional homogeneous or heterogeneous materials are difficult to work. In this paper authors have applied their SACs for the hydrogenation of nitroarenes, a well-established reaction for which a number of base metal heterogeneous catalysts are reported. Previously reported catalysts worked more efficiently and exhibited high chemoselectivity with very broad substrate scope compared to these Co-SACs. Also, the reported ones work under very milder conditions (<40 oC and <5 bar) compared to these Co-SACs, these require drastic conditions (110-120 oC and 20-30 bar H₂). Hence the applicability of these Co-SACs catalyst is not interesting and potential for the hydrogenation of nitro compounds. In recent years, a number of single atoms catalysts and methodologies to prepare them have been well established and reported. The development and applications of single atoms, which bridge homogeneous and heterogeneous catalysis aspects is interesting, if the new SACs catalysts work for challenging reactions for which the homogeneous complexes or heterogeneous materials are difficult to work. However, this is not the case in this reported paper, the new Co-SACs are applied for very simple reaction, the hydrogenation of nitro arenes. This paper is not up to the mark to publish in Nature Communication.

Responses to Editor and Reviewers

Dear Editor and Reviewers,

Thank you very much for your valuable comments and suggestions that greatly helped to improve our manuscript. In revising the manuscript we have carefully considered your comments and suggestions, which have been taken into full account in the revised manuscript.

Reviewer 1#

The authors reported in this work a single-atom catalyst with $\text{Co}_1\text{-N}_3\text{P}_1$ sites that exhibits very high activity and selectivity for hydrogenation of nitroarenes. The experiments were nicely designed and very well done. The computational part, however, is not convincing. The following issues need to be addressed before I can recommend accepting the manuscript.

Question 1. The reaction path shown in Fig. 4 is only a sketch that is based on the previously proposed mechanism of homogeneous catalysis. It has been well known that the heterogeneous catalysis can be very different from the homogeneous one. It's not to just 'borrow' the mechanism of homogeneous catalysis here without any support from either experiments or calculations. This is the biggest problem of the paper.

Answer: Thank you for your comment. Indeed heterogeneous catalysis is often quite different from homogeneous catalysis. However, considering that the coordination structure of Co single-atom catalyst is similar to that in the homogeneous catalysts, we further performed DFT calculations to explore the mechanism of hydrogenation of nitrobenzene over Co_1/NPC catalyst with similar way reported in literatures. As shown in Fig. 5 and Supplementary Fig. 23, one H_2 molecule first goes through heterolytic cleavage with assistance of H_2O -mediated H-shuttling mechanism to form $\text{Co-H}^{\delta-}$ and $\text{P-H}^{\delta+}$ on the $\text{Co}_1\text{-N}_3\text{P}_1$ sites, which can serve as the initial state for hydrogenation process. Then, the target nitrobenzene molecule was adsorbed on $\text{Co}_1\text{-N}_3\text{P}_1$ site with a free energy of -0.78 eV (Fig. 5b, I). Subsequently, the activated H atom on $\text{Co-H}^{\delta-}$ and the O atom of PhNO_2 are combined to produce PhNOOH intermediate, which is later reduced to PhNO intermediate by the H atom transfer from $\text{P-H}^{\delta+}$ (II). Notably, the PhNO intermediate can also be detected during the reaction process (Supplementary Fig. 25). After that, another H_2 molecule is dissociated to form activated H atom, which attacks the oxygen atom of PhNO and reduces it to PhNHOH (III, IV). It is worth noting that the adsorption energies of intermediates PhNO and PhNHOH on $\text{Co}_1\text{-N}_3\text{P}_1$ site are more favorable than Ph-NO_2 , ensuring the reaction progress of

targeted substrate (Supplementary Fig. 26). In the next step, the third H₂ molecule participates in and the formed H atom interacts with PhNHOH to generate the final PhNH₂ product (V, FS). It can be seen the whole process is highly endothermic, confirming the possibility of proposed reaction path.

Above discussions were added to the revised manuscript.

Fig. 5 Reaction mechanism of the nitrobenzene reduction. (a) The proposed reaction pathway for the hydrogenation of nitrobenzene to aniline at Co-P interfacial site. (b) Energy profile of hydrogenation of nitrobenzene over Co₁-N₃P₁ site.

Question 2. Increase in electron density of Co is a selling point of the paper. The Bader charge shall be provided to support this.

Answer: Thank you for your thoughtful suggestion. We have performed bader charge analysis. The charge state of Co₁/NPC was estimated to be +0.81 e, while that of Co₁/NC was +0.97 e, indicating an obviously decrease of oxidation state and increased electron density of Co in Co₁-N₃P₁ site. The oxidation state of Co in Co₁/NC is predicted to be +2 having the low spin with (d_{z2})²(d_{x2-y2})²(d_{yz})² (d_{xz})¹ (d_{xy})⁰ configuration in the quasi-planar field, while the oxidation state of Co in Co₁/NPC is predicted to be +1 having (d_{z2})²(d_{x2-y2})²(d_{yz})² (d_{xz})² (d_{xy})⁰ configuration in the

quasi-planar field. The Bader analysis of Co in both Co₁/NPC and Co₁/NC agrees well with the XANES result.

Above discussions were added to the revised manuscript.

Question 3. It was claimed in the paper that H₂ heterolytic cleavage will form metal-H^{δ-} and heteroatom-H^{δ+}, but the Bader charges were not provided. Adsorption energy of individual H atoms after cleave can also be provided which will be helpful in explaining the proposed mechanism.

Answer: Following your suggestion, we have performed Bader charge analysis. The charge state of Co₁-N₃P₁ site was estimated to be +0.81 *e*, whereas the Co₁-N₄ site was +0.97 *e*, indicating an obviously decrease of oxidation state of Co in Co₁-N₃P₁ site, which is consistent with the XANES results.

In addition, we studied the H₂ adsorption and activation on Co₁/NPC and Co₁/NC catalysts to investigate the adsorption energy of individual H atoms after cleave (Supplementary Fig. 20). H₂ molecule is adsorbed on Co atom of Co₁/NPC with adsorption energy of -0.08 eV, and the bond length of H₂ is 0.814 Å, which is much longer than free molecular H₂ (0.752 Å) in gas phase. While H₂ molecule is adsorbed on Co atom of Co₁/NC with adsorption energy of 0.033 eV, and the bond length of H₂ is 0.783 Å. The much longer H-H bond length of adsorbed H₂ indicates that Co₁/NPC has higher activation than Co₁/NC. After dissociation, the adsorption energy of H atoms on Co₁/NC is 1.11 eV, while is -0.20 eV on Co₁/NPC.

Supplementary Fig. 20 The initial states and final states for the direct dissociation of hydrogen over (a-b) Co₁/NC and (c-d) Co₁/NPC in the absence of water.

Reviewer 2#

In this work, Song et al. reported an unsymmetrically coordinated single atomic catalyst with $\text{Co}_1\text{-N}_3\text{P}_1$ sites for hydrogenation of functionalized nitroarenes. This is an excellent example of enhancing catalytic activity by modifying the coordination structure of single atom catalysts. The catalyst in this work shows impressive activity and selectivity in nitrobenzene hydrogenation reaction. The mechanism of the catalytic reaction was also thoroughly investigated by kinetic isotope effect and density functional theory calculations. In my opinion, this is an excellent research work and it can be published in Nature Communications after minor revision. These issues need to be addressed before publication:

Question 1. Freeze-drying is an important step in the preparation process of Co_1/NPC catalyst and is suggested to be involved in Scheme S1. In addition, there is a typo in line 266, where "before" should be "after".

Answer: We have updated the Scheme S1 in the revised Supporting Information, and corrected the mistake in the revised manuscript.

Scheme S1. Schematic illustration of the preparation process for Co_1/NPC catalyst.

Question 2. It is difficult to distinguish the positions of absorption edges in Fig. 1d. It is suggested to enlarge the details or add a first derivative diagram in the supplementary information to show the differences.

Answer: Thank you for your thoughtful suggestion. The XANES spectra was enlarged as shown in the inset of Fig. 1d. It can be seen that the edge position of Co_1/NPC is more negative than that of Co_1/NC , and the white line of Co_1/NPC is weaker than that of Co_1/NC , indicating the Co single atoms in Co_1/NPC carry more charge. Such difference could be attributed to the less electron transfer from Co to support in Co_1/NPC than that in Co_1/NC due to the electron donation of P atom, that is, the electron can transfer from P to Co because of the weaker electronegativity of P than N atoms (Ref: J. Am. Chem. Soc. 2020, 142, 8431).

Fig. 1 (d) Co K-edge XANES spectra.

Question 3. The scale bar of Supplementary Fig. 3 is not clearly visible.

Answer: We have added the scale bar in the revised Supplementary Information.

Question 4. Other characterizations about N, P dual-coordination should be provided to further support the single site structure, such as EELS.

Fig. 2 Atomic resolution EELS analysis of Co₁/NPC sample. (a) AC STEM-HAADF image of Co₁/NPC sample at a relatively low beam current. (b) HAADF image acquired simultaneously with atomic resolution EELS mapping of Co₁/NPC at the red square area in (a). (c) The overlap map of P, N and Co elements. (d-f) Elemental maps of P, N, Co, respectively. (g-i) The spectra extracted at the yellow circled position from (d), (e), (f), respectively, which proves the existence of the three elements.

Answer: Thank you for your suggestion. We further performed atomic resolution electron energy loss spectroscopy (EELS) at a relatively low beam current to

minimize electron-beam perturbations to provide a strong evidence of $\text{Co}_1\text{-N}_3\text{P}_1$ structure (Fig. 2a-b). The extracted Co L-edge EELS spectrum from Fig. 2d presents a clear Co signal (Fig. 2g), providing direct evidence for the presence of atomically dispersed Co species. More importantly, the existence of N, P dual-coordination vicinal to Co site was revealed by identifying the surrounding heteroatoms. Fortunately, from the N, P and overlap maps (Fig. 2e-f, 2c), it clearly exhibits that three N atoms (green) and one P atom (red) exist around the Co site. Atomic-scale N K-edge and P L-edge EELS spectra collected at the corresponding positions from Fig. 2e and Fig. 2f are further demonstrated the N and P signals (Fig. 2h-i). This forcefully confirms the $\text{Co}_1\text{-N}_3\text{P}_1$ configuration in Co_1/NPC sample.

Question 5. According to the proposed reaction mechanism in Figure 4, the adsorption energy of intermediates Ph-NO and Ph-NHOH on Co-H^δ should be more favorable than Ph- NO_2 . DFT calculations are suggested to confirm this.

Answer: Following your suggestion, we have carried out DFT calculations to compare the adsorption energies of intermediates. As shown in Supplementary Fig. 26, the adsorption energies of Ph- NO_2 , Ph-NHOH and Ph-NO on Co_1/NPC were estimated to be -0.64 eV, -0.77 and -0.92 eV, respectively. This result indicates the intermediates Ph-NHOH and Ph-NO were more favorable adsorbed than Ph- NO_2 on the catalyst surface, and thus, ensuring the complete conversion.

Supplementary Fig. 26 The calculated adsorption energies of Ph- NO_2 , Ph-NHOH and Ph-NO on $\text{Co}_1\text{-N}_3\text{P}_1$ interface site.

Question 6. During the kinetic isotope effect study, because H_2O is also used as co-solvents and H/D exchange between D_2 and H_2O is facile. Thus, D_2 and D_2O should be noted specially in the manuscript.

Answer: We have added the detailed information into the caption of Fig. 4a-b.

Question 7. I notice there is a little activity decline in the re-used experiments in supplementary Fig. 20. The authors should comment on the reasons for activity loss.

Answer: Thank you for your suggestion. A slight decrease of activity is observed after five cycles with Co_1/NPC , which can be ascribed to the loss of some

catalysts and active Co species during the recycling experiments (Supplementary Table 1).

Reviewer 3#

Song and co-authors report the synthesis of Co single atom catalyst with unsymmetrical $\text{Co}_1\text{-N}_3\text{P}_1$ active site that displays high activity and chemoselectivity for hydrogenation of nitroarenes. Attractive catalytic performance is attributed to the increased electron density of P-modulated $\text{Co}_1\text{-N}_3\text{P}_1$ active site, which facilitate H_2 dissociation as verified by kinetic isotope effect and density functional theory calculations. These interesting findings make an important contribution to the field of single-atom catalysis. Before the publication, addressing following issues can further solidify this work.

Question 1. The authors need to better explain why only one N atom has been replaced other than two or more?

Answer: Thank you for your suggestion. First, we conducted the quantitative least-squares EXAFS curve-fitting for the $\text{Co}_1\text{/NPC}$ sample in Fig. 1e, where the best-fitting analysis displayed that the main Co-N first-shell coordination was 3.2 ± 0.1 and the shoulder Co-P coordination was 0.9 ± 0.1 , approximate the $\text{Co}_1\text{-N}_3\text{P}_1$ structure (Supplementary Table 2).

In order to better clarify this structure, we further obtained the objective structure information of $\text{Co}_1\text{-N}_3\text{P}_1$ site at the atomic level with the help of electron energy loss spectroscopy (EELS). As shown in Fig. 2a-b, the extracted EELS spectrum at the yellow circled position from Fig. 2d in the Co map, which is nearly the size as a single atom, presents a clear Co signal (Fig. 2g), providing direct evidence for the presence of atomically dispersed Co species. More importantly, the presences of N, P dual-coordination vicinal to Co site were revealed by identifying four heteroatoms (N and P) on the adjacent sides. Fortunately, from the N, P and overlap maps (Fig. 2c, 2e-f), it clearly exhibits that three N atoms and one P atom exist around the Co site. Atomic-scale N K-edge and P L-edge EELS analysis extracted at the yellow circled position from Fig. 2e and Fig. 2f are further demonstrated the N and P signals (Fig. 2h-i). Based on above experimental investigations, it can be deduced that the single Co atom in $\text{Co}_1\text{/NPC}$ sample is anchored by three N atoms and one P atom, i.e., $\text{Co}_1\text{-N}_3\text{P}_1$ structure.

Furthermore, we performed the DFT calculations to investigate the formation energies with different coordination environments (Supplementary Fig. 16). It can be seen that the $\text{Co}_1\text{-N}_3\text{P}_1$ structure possesses the most favorable formation energy (-0.864 eV) among other various structures, such as $\text{Co}_1\text{-N}_1\text{P}_3$ and $\text{Co}_1\text{-N}_2\text{P}_2$. All above results suggest the highly reliable $\text{Co}_1\text{-N}_3\text{P}_1$ coordination structure.

Fig. 2 Atomic resolution EELS analysis of Co₁/NPC sample. (a) AC STEM-HAADF image of Co₁/NPC sample at a relatively low beam current. (b) HAADF image acquired simultaneously with atomic resolution EELS mapping of Co₁/NPC at the red square area in (a). (c) The overlap map of P, N and Co elements. (d-f) Elemental maps of P, N, Co, respectively. (g-i) The spectra extracted at the yellow circled position from b, c, d, respectively, which proves the existence of the three elements.

Supplementary Fig. 16 Various structures and formation energies of Co_1/NPC sample.

Question 2. The authors comment on whether the introduction of phosphonic acid affects the surface density of active sites of CoN_3P as compared to CoN_4 .

Answer: We have performed the FT-IR analysis of Co_1/NPC (Supplementary Fig. 1). It can be seen that the $-\text{PO}_3^-$ functional group of precursor AePA (phosphonic acid) was absent after pyrolyzation, indicating P species were doped into carbon matrix. Thus, the introduced phosphonic acid only plays the role of anchoring Co single-atom sites.

Supplementary Fig. 1 FTIR spectra of AePA and Co_1/NPC sample.

Question 3. How is the amplitude reduction factor S_0^2 determined? There are

different descriptions in main text (0.712) and SI (0.87/0.90). EXAFS data in K space need to be given.

Answer: Thank you for your carefully reading. We are sorry for the different description of S_0^2 . We corrected it in the revised manuscript. S_0^2 was set as 0.87 for Co-N, which was obtained from the experimental EXAFS fit of reference CoO by fixing CN as the known crystallographic value and was fixed to all the samples. Additionally, the EXAFS data in K space of Co₁/NC and Co₁/NPC was added to the revised Supplementary Information (Supplementary Fig. 14).

Supplementary Fig. 14 EXAFS fitting analysis of Co₁/NC and Co₁/NPC in k space.

Question 4. The authors need to better present the curves in Figure 1.d to show edge positions.

Answer: Following your suggestion, the XANES spectra was enlarged as shown in the inset of Fig. 1d. It can be seen that the edge position of Co₁/NPC is more negative than that of Co₁/NC, and the white line of Co₁/NPC is weaker than that of Co₁/NC, indicating the Co single atoms in Co₁/NPC carry more charge. Such difference could be attributed to the less electron transfer from Co to support in Co₁/NPC than that in Co₁/NC due to the electron donation of P atom, that is, the electron can transfer from P to Co because of the weaker electronegativity of P than N atoms (Ref: J. Am. Chem. Soc. 2020, 142, 8431).

Fig. 1 (d) Co K-edge XANES spectra.

Question 5. Please explain why there is no shoulder feature around 7717 eV for the XANES of sample Co₁/NC containing Co-N₄ configuration.

Answer: Ideally, Co single atom anchored in perfect uniform graphene or phthalocyanine support with in-plane Co-N₄ configuration has a D_{4h} symmetry shoulder around 7717 eV in the XANES spectrum. However, when the support has many defects, the shoulder feature becomes invisible. Indeed, two characteristic peaks of carbon at 1344 cm⁻¹ (D band, disordered/defective carbon) and 1576 cm⁻¹ (G band, graphitic carbon) were detected in the Raman spectra of Co₁/NC and Co₁/NPC (Supplementary Fig. 2). The intensity ratios (I_D/I_G) of Co₁/NC and Co₁/NPC were ca. 1.07 and 1.09, respectively, indicating the carbon matrices in both two samples were disordered with a large number of defects, that is, our support is not the perfect in-plane CoPc structure. This will result in the distorted structure, and thus the D_{4h} symmetry cannot be observed in Co₁/NC sample. Such phenomenon was also frequently reported in other literature.

Supplementary Fig. 2 Raman spectra of Co₁/NC and Co₁/NPC.

Question 6. Please explain why the white line of XANES of sample Co₁/N₃P is

sharper than that of Co₁/NC. Introducing Co-P bonding could result in the red shift of the features in XANES?

Answer: Thank you for your carefully observation. We re-normalized the XANES curves and the correct result was shown in Fig. 1d. It can be seen that the white line of XANES of sample Co₁/NPC is weaker than that of Co₁/NC, and the edge position of Co₁/NPC is more negative than that of Co₁/NC, indicating the Co single sites in Co₁/NPC sample carry more charge. Such difference could be attributed to the less electron transfer from Co to P because of the weaker electronegativity of P than N atoms.

Fig.1 (d) Co K-edge XANES spectra.

Question 7. As the authors claim the importance of the local structure, it is highly recommended to supply the theoretical XANES simulation to further prove the proposed CoN₃P and CoN₄ structure.

Answer: Following your suggestion, and to better prove the proposed CoN₃P and CoN₄ structure, we have conducted the simulation to supply the theoretical XANES data. The Co K-edge theoretical XANES simulations were carried out with the FDMNES code in the framework of real-space full multiple-scattering (FMS) scheme using Muffin-tin approximation for the potential. The simulated XANES curves of the proposed structures are in a good agreement with experimental XANES data (Fig. 1f, Supplementary Fig. 15), indicating the rationality of two structures.

Fig. 1 (f) The experimental XANES curve in comparison with the calculated XANES data of $\text{Co}_1\text{-N}_3\text{P}_1$ site in Co_1/NPC sample. Inset: the schematic atomic structure of Co_1/NPC derived from the EXAFS results.

Supplementary Fig. 15 The experimental XANES curve in comparison with the calculated XANES data of $\text{Co}_1\text{-N}_4$ site in Co_1/NC sample. Inset: the schematic atomic structure of Co_1/NC derived from the EXAFS results.

Question 8. Keep abbreviations always consistent, for example ‘Fourier transformed EXAFS’ in line 98 and ‘FT-EXAFS’ in line 99.

Answer: We have carefully checked and corrected these inconsistencies in our revised manuscript.

Question 9. In DFT calculations part, the representation of atomic orbital should be unified.

Answer: We have corrected the atomic orbital in our revised manuscript.

Question 10. The authors should provide more details about the calculated model and parameters, such as lattice parameter of model, KPOINTS, the formula of binding energy, the frequency of IS (TS and FS).

Answer: Thank you for your suggestion. We have provided the details about the calculated model and parameters in our revised manuscript.

Question 11. A number of relevant references should be cited, including Co SACs on graphene for hydrogenation of nitroarenes, tunable local structures of Co SACs and review papers related to local atomic environments of single atoms catalysis etc.

Answer: Thank you for your suggestion. We cited the relevant references in the revised manuscript as ref. 11,, 14-15, 17-18, 31-40.

Reviewer 4#

This paper reports the preparation of cobalt single atom catalysts and their application for the hydrogenation of nitro compounds to anilines. In recent years, the development of single atom catalysts increases the interests of researchers. In fact, the search of new catalytic materials those should perform challenging reactions in more efficient and selective way, for those the traditional homogeneous or heterogeneous materials are difficult to work. In this paper authors have applied their SACs for the hydrogenation of nitroarenes, a well-established reaction for which a number of base metal heterogeneous catalysts are reported. Previously reported catalysts worked more efficiently and exhibited high chemoselectivity with very broad substrate scope compared to these Co-SACs. Also, the reported ones work under very milder conditions (<40 °C and <5 bar) compared to these Co-SACs, these require drastic conditions (110-120 °C and 20-30 bar H₂). Hence the applicability of these Co-SACs catalyst is not interesting and potential for the hydrogenation of nitro compounds. In recent years, a number of signal atoms catalysts and methodologies to prepare them have been well established and reported. The development and applications of single atoms, which bridge homogeneous and heterogeneous catalysis aspects is interesting, if the new SACs catalysts work for challenging reactions for which the homogeneous complexes or heterogeneous materials are difficult to work. However, this is not the case in this reported paper, the new Co-SACs are applied for very simple reaction, the hydrogenation of nitroarenes. This paper is not up to the mark to publish in Nature Communication.

Answer: Indeed, recent years have witnessed the rapid development of single atom catalysts in many different kinds of catalytic reactions. And as we known, modulating the atomic coordination structure has emerged as a promising strategy to further improve the catalytic performance of SACs. As for hydrogenation of nitro compounds, a very important in fine chemical and pharmaceutical industry to produce amines, all the reported SACs were based on

metal₁-N_x-C catalysts. For SACs with M₁-N_x-C sites, the symmetric electronic distribution may limit the activation of reactants, thereby leading to the hampered catalytic kinetics and performances. Recent studies have found that introducing heteroatom P for an unsymmetrical N/P mixed-coordination can further modulate the electronic properties of center metal atoms. The unsymmetrical geometric structure can evoke the distortion of electronic density and alter the d-band center. However, there is no related report in hydrogenation of nitroarenes. Thus, we anticipated that constructing the unsymmetrical N/P dual-coordinated transition metal SACs would improve the catalytic performance for hydrogenation of nitroarenes. This is the starting point of our work. In our opinion, the novelty of this work contains the following points:

(1) We developed an efficient route to prepare Co SAC with particular Co₁-N₃P₁ coordination structure. Especially, besides XAFS analysis, the coordination structure was first directly observed through atomic resolution EELS analysis, providing tough evidence of the proposed configuration.

(2) The prepared Co₁/NPC catalyst with Co₁-N₃P₁ coordination structure exhibited 60-fold higher activity than that of traditional Co₁-N₄ SAC and one order of magnitude higher than the state-of-the-art M₁-N_x-C SACs in literatures. As reviewer's suggestion, we found the reference about the hydrogenation of nitrobenzene under mild reaction condition (40 °C, 1 bar H₂) (Ref: Zhou et al. Sci. Adv. 2017; 3: e1601945). The reported results showed a high nitrobenzene conversion of 98.7% within 18 hours. Thus, we also evaluated the catalytic performance over our Co₁-N₃P₁ sample under similar reaction condition. From Supplementary Fig. 18, It can be seen that Co₁-N₃P₁ also exhibits much higher activity than the result reported by Zhou et al.

Supplementary Fig. 18 Time course of nitrobenzene conversions over Co₁/NPC sample. Reaction conditions: 22 mg catalyst, 0.5 mmol nitrobenzene, 40 mL EtOH/H₂O (v:v=4:1), 40 °C, 1 bar H₂.

(3) We clarified the intrinsic reason for high activity of Co₁-N₃P₁ and the reaction

mechanism for hydrogenation of nitrobenzene through thoroughly experiments and DFT calculations. The findings in this work provide a new insight into the rationally modulating the coordination structure of central metal atoms for boosting catalytic performance of SACs in heterogeneous catalysis.

REVIEWERS' COMMENTS

Reviewer #1 (Remarks to the Author):

The authors nicely addressed all issues I raised in the previous round of review. I recommend accepting the manuscript as it is.

Reviewer #2 (Remarks to the Author):

The authors have properly addressed my previous comments and revised the manuscript accordingly. No further concerns will be raised by this reviewer.

Reviewer #3 (Remarks to the Author):

The authors have made great effort in revision. The manuscript is significantly improved and becomes stronger. I recommend the publication after the authors include these relevant references.

Nature communications 9 (1), 1-9, 2018

ACS Catalysis 10 (10), 5862-5870, 2020

Advanced Materials, 2008471, 2021

Responses to Editor and Reviewers

Dear Editor and Reviewers,

Thank you very much for your valuable comments and suggestions that greatly helped to improve our manuscript. In revising the manuscript we have carefully considered your comments and suggestions, which have been taken into full account in the revised manuscript.

Reviewer 3#

Comments: The authors have made great effort in revision. The manuscript is significantly improved and becomes stronger. I recommend the publication after the authors include these relevant references. Nature communications 9 (1), 1-9, 2018; ACS Catalysis 10 (10), 5862-5870, 2020; Advanced Materials, 2008471, 2021.12.

Answer: Thank you for your suggestion. We have cited the above references in the revised manuscript as refs. 4, 29-30.